# C Type Natriuretic Peptide Receptor Activation Inhibits Sodium Channel Activity in Human Aortic Endothelial Cells by Activating the Diacylglycerol-Protein Kinase C Pathway

**DOI:** 10.3390/ijms232213959

**Published:** 2022-11-12

**Authors:** Ling Yu, Mohammad-Zaman Nouri, Lauren P. Liu, Niharika Bala, Nancy D. Denslow, John F. LaDisa, Abdel A. Alli

**Affiliations:** 1Department of Physiology and Aging, College of Medicine, University of Florida, Gainesville, FL 32610, USA; 2Department of Physiological Sciences and Center for Environmental and Human Toxicology, University of Florida, Gainesville, FL 32610, USA; 3Department of Pediatrics, Section of Cardiology, Medical College of Wisconsin, Milwaukee, WI 53226, USA; 4The Herma Heart Institute, Children’s Wisconsin, Milwaukee, WI 53226, USA; 5Department of Biomedical Engineering, Marquette University and the Medical College of Wisconsin, Milwaukee, WI 53226, USA; 6Department of Medicine, Division of Cardiovascular Medicine, Medical College of Wisconsin, Milwaukee, WI 53226, USA; 7Department of Physiology, Medical College of Wisconsin, Milwaukee, WI 53226, USA; 8Department of Medicine, Division of Nephrology, Hypertension, and Renal Transplantation, College of Medicine, University of Florida, Gainesville, FL 32610, USA

**Keywords:** natriuretic peptide receptor C, endothelial sodium channels, diacylglycerols, protein kinase C, aortic endothelial cells

## Abstract

The C-type natriuretic peptide receptor (NPRC) is expressed in many cell types and binds all natriuretic peptides with high affinity. Ligand binding results in the activation or inhibition of various intracellular signaling pathways. Although NPRC ligand binding has been shown to regulate various ion channels, the regulation of endothelial sodium channel (EnNaC) activity by NPRC activation has not been studied. The objective of this study was to investigate mechanisms of EnNaC regulation associated with NPRC activation in human aortic endothelial cells (hAoEC). EnNaC protein expression and activity was attenuated after treating hAoEC with the NPRC agonist cANF compared to vehicle, as demonstrated by Western blotting and patch clamping studies, respectively. NPRC knockdown studies using siRNA’s corroborated the specificity of EnNaC regulation by NPRC activation mediated by ligand binding. The concentration of multiple diacylglycerols (DAG) and the activity of protein kinase C (PKC) was augmented after treating hAoEC with cANF compared to vehicle, suggesting EnNaC activity is down-regulated upon NPRC ligand binding in a DAG-PKC dependent manner. The reciprocal cross-talk between NPRC activation and EnNaC inhibition represents a feedback mechanism that presumably is involved in the regulation of endothelial function and aortic stiffness.

## 1. Introduction

In addition to clearing natriuretic peptides from circulation, the C type natriuretic peptide receptor (NPRC) is coupled to multiple intracellular signaling pathways [1]. NPRC activation is linked to the inhibition of the adenylyl cyclase pathway [2], activation of the phospholipase C pathway [3], activation of the nitric oxide system [4] and arachidonic acid induced mobilization of intracellular calcium [5]. NPRC is a disulfide linked homodimer that binds all natriuretic peptides with high affinity. NPRC is expressed in multiple cell types including endothelial cells [6,7], vascular smooth muscle cells [8], gastric muscle cells [9], glomerular epithelial cells [10], mesangial cells [11,12], pancreatic alpha cells [13], renal epithelial cells [14], and various types of cancer cells [15,16,17,18,19]. This cell surface receptor is phosphorylated [20] and glycosylated [21] at multiple sites. NPRC is regulated by microRNAs [22], protein kinases [23,24,25], prostaglandins [26], β-adrenergic stimulation [26], growth factors [7], and cytokines [25].

Endothelial sodium channels (EnNaC) are expressed in vascular smooth muscle and in the endothelium [27]. These channels were shown to play an important role in aldosterone-dependent endothelial stiffness. EnNaC has been shown to be mechano-sensitive and over-activation of these channels have been suggested to play a role in endothelium dysfunction [28]. Although EnNaC activity is thought to be positively regulated by phosphatidylinositol bisphosphates (PIP2) the role of NPRC in the PLC-dependent hydrolysis of these negatively charged lipids and EnNaC regulation has not been investigated

In this study, we tested our hypothesis that NPRC activation by ligand binding inhibits the function of EnNaC in human aortic endothelial cells in a mechanism involving the DAG-PKC pathway. The goal of this study was to (i) determine whether NPRC activation regulates EnNaC protein expression and activity in human aortic endothelial cells (hAoEC) and (ii) investigate changes in plasma membrane diacylglycerol (DAG) and protein kinase C (PKC) activity in these cells after cANF mediated activation of NPRC in order to elucidate the mechanism by which NPRC can regulate EnNaC.

## 2. Results

### 2.1. NPRC Activation in Response to cANF Treatment Reduces EnNaC Alpha Subunit Protein Expression in hAoECs

Here, we focused our studies on the alpha subunit of EnNaC in hAoECs. hAoECs were treated with either vehicle or cANF for 8 h before harvesting the cells for protein and then Western blotting for EnNaC alpha subunit. Densitometric analysis of the 75 kDa and 60 kDa immunoreactive bands in Western blots shows cANF mediated activation of NPRC attenuates EnNaC protein expression in hAoEC (Figure 1).

### 2.2. EnNaC Activity Decreases in hAoECs after cANF Mediated Activation of NPRC

Next, we performed patch clamp experiments to further corroborate the Western blot studies and show NPRC activation by cANF negatively regulates EnNaC at the level of channel activity (NPo). As shown in Figure 2, cANF mediated activation of NPRC reduced the activity (NPo) of EnNaC with a conductance of 25 pS.

### 2.3. siRNA Dependent Knockdown of NPRC Blunts the Decrease of EnNaC Alpha Subunit Protein Expression after cANF Treatment in hAoECs

We performed cell surface biotinylation and Western blot studies in order to investigate whether the density of EnNaC protein in plasma membrane fractions is altered after treatment with cANF compared to control each for 8 h. In order to corroborate the role of NPRC activation by ligand binding in the reduction of EnNaC protein expression, we transfected three separate batches of hAoEC with siRNA targeting NPRC or non-targeting control siRNA for 72 h before harvesting the cells for protein and Western blotting for EnNaC alpha protein expression. The activation of NPRC with the agonist cANF decreased EnNaC alpha subunit protein expression in hAoEC, but this was blunted in cells transfected with NPRC specific siRNA compared to cell transfected with non-targeting control siRNA (Figure 3).

### 2.4. cANF Mediated Activation of NPRC Increases DAG Abundance in hAoECs

The activation of the phospholipase C (PLC) pathway upon NPRC ligand binding has not been investigated in hAoECs. Here, we investigated changes in DAGs after treatment of hAoEC with the NPRC agonist cANF or vehicle for 8 h. Multiple diacylglycerols including DAG(18:1/20.2) and DAG(16:1/18:3) increased in hAoECs treated with cANF compared to vehicle (Figure 4).

### 2.5. PKC Activity Is Elevated in hAoEC Treated with cANF Compared to Cells Treated with Vehicle

In order to determine whether NPRC activation inhibits EnNaC protein expression and activity in hAoEC by the activation of PKC, we measured PKC activity in cells treated with vehicle or cANF. Treatment of hAoEC with the NPRC agonist cANF resulted in an increase in PKC activity when compared to cells treated with vehicle (Figure 5).

### 2.6. Inhibition of PKC Activity Augments EnNaC Protein Expression in hAoEC

Next, we attempted to further investigate the role of PKC in the cross-talk between NPRC and EnNaC in hAoEC. As shown in Figure 6, pharmacological inhibition of PKC resulted in a significant increase in EnNaC protein expression and this effect was blunted with cANF treatment.

## 3. Discussion

In this study we investigated the regulation of EnNaC expression and activity after NPRC ligand binding in hAoECs. We previously showed EnNaC [29] and NPRC [30] proteins are both expressed in these cells, but this is the first investigation of a cross-talk between EnNaC and NPRC. In addition, we showed an elevation of various DAGs in hAoECs after activation of NPRC by the agonist cANF. We further investigated whether the increase in various DAGs upon activation of NPRC increases PKC activity in hAoEC.

Multiple studies have investigated the role of NPRC signaling in the regulation of current. Rose et al. showed NPRC activates transient receptor potential channel transcripts in cardiac fibroblasts [31]. This group showed cANF and CNP mediated effects were similar to that of the DAG analogue 1-oleoyl-2-acetyl-sn-glycerol (OAG). Another study suggested a role for NPRC in the inhibition of L-type calcium current in a mechanism involving activation of inhibitory G protein alpha subunits and subsequent inhibition of adenylyl cyclase [32]. Simon et al. showed the NPRC agonist cANF causes endothelial cell hyperpolarization through the activation of chloride channels [33]. However, the role of NPRC activation in the regulation of EnNaC in hAoEC has never been investigated.

Our current study was focused on the regulation of the alpha subunit of EnNaC since this subunit has been shown to be sufficient for channel activity [34] and this subunit could alter endothelial stiffness by itself [35]. Our patch clamp studies showed a decrease in EnNaC activity in hAoEC treated with cANF compared to vehicle (Figure 2). Additionally, we showed a 60 kDa immunoreactive band, presumably corresponding to the cleaved form of EnNaC, is decreased in hAoEC treated with cANF compared to vehicle (Figure 3).

Published studies have demonstrated an increase in endothelial cell stiffness results from the activation of EnNaC. For example, Zhang et al. showed EnNaC activation by high salt and endothelial cell mineralocorticoid receptor activation augments aortic endothelium stiffness in mice [36]. Consistent with these results, Xiong et al. showed EnNaC contributes to renal artery endothelial stiffening [37]. Accordingly, Martinez-Lemus et al. showed that the inhibition of EnNaC with amiloride significantly reduces the development of endothelial and aortic stiffness [38]. To our knowledge, this is the first study to investigate the regulation of EnNaC by NPRC activation, although we did not directly measure how this cross-talk affects endothelial stiffness. The suppression of EnNaC activity by ligand binding of NPRC presumably mitigates cell stiffness in aortic endothelial cells in a mechanism independent of dietary salt. Instead, our data suggest the activation of NPRC reduces EnNaC proteolysis which would in turn inhibit EnNaC activity (Figure 3).

The hydrolysis of PIP2 at the cell membrane results in the release of inositol 1, 4, 5 trisphosphate (IP3) into the cytoplasm and the diffusion of DAG through the membrane. DAG is an important lipid signaling messenger that increases cardiomyocyte contraction [39] through the activation of PKC. Here, we show various forms of DAG (Figure 4) as well as PKC activity (Figure 5) is augmented in hAoEC treated with the NPRC agonist cANF compared to vehicle. PKC is known to inhibit renal ENaC activity [40], and presumably, EnNaC is inhibited by a similar mechanism in hAoEC. Here, we show pharmacological inhibition of PKC augments EnNaC protein levels in human aortic endothelial cells and this effect is partially blocked by the NPRC agonist cANF (Figure 6). The activation of PKC in hAoEC has been previously shown to result in increased production and release of c-type natriuretic peptide (CNP) in hAoEC [41]. Whether CNP production following NPRC activation further inhibits EnNaC activity in these cells remains to be investigated.

One limitation of this study is that we did not investigate whether post-translational modifications of NPRC including phosphorylation and glycosylation are involved in the activation of the receptor and regulation of EnNaC in hAoECs. A second limitation is that we did not investigate whether the increased production of various DAGs resulting from cANF mediated activation of NPRC directly or indirectly inhibit EnNaC protein expression and activity in these cells. In renal epithelial cells, DAGs play a role in the activation of PKC which negatively regulates MARCKS and ENaC. The regulation of EnNaC by MARCKS in a PKC dependent manner in hAoECs has not been investigated. Finally, we did not investigate how the inhibition of EnNaC activity following the activation of NPRC affects endothelial cell stiffnesses. Additional studies will be performed to address these knowledge gaps.

In conclusion, our data show for the first time NPRC activation by ligand binding attenuates EnNaC protein expression and activity in human aortic endothelial cells (Figure 7). We also show an increase in DAG and PKC activity by cANF treatment plays a role in the crosstalk between NPRC and EnNaC in human aortic endothelial cells (Figure 7).

## 4. Materials and Methods

### 4.1. Reagents

Human cANF was purchased from Phoenix Pharmaceuticals Inc. (Mountain View, CA, USA). GF109203X was purchased from (Sigma; St. Louis, MO, USA). All other reagents were purchased from Thermo Fisher Scientific.

### 4.2. Cell Culture

Human aortic endothelial cells were purchased from Cell Biologics (Chicago, IL, USA) and cultured in complete growth media (Cell Biologics; Chicago, IL, USA) in a humidified incubator set to 5% CO_2_, 37 °C. Cells with a passage number of less than three were used for experiments.

### 4.3. Electrophysiology

Single channel patch clamp recordings for highly selective sodium channels were performed as previously described by our lab [42] with the following modifications. Briefly, micropipettes were made using filamented borosilicate glass 130 capillaries (TW-150F, World Precision Instruments, Sarasota, FL, USA) using a two-stage vertical puller (Narishige, Tokyo, Japan) and had a resistance of 6–10 MΩ. HAoEC were subcultured on 12 mm permeable transwell inserts. Cells were treated with the NPR-C agonist, cANF, or vehicle for 30 min before starting to patch and the treated cells were patched for up to 3 h. An I-V curve as plotted after different voltages were applied to patched cells. Channel activity in a patch was calculated using pCLAMP software (Molecular Devices, Sunnyvale, CA, USA)

### 4.4. siRNA Mediated Knockdown of NPRC

HAoEC were transfected with siGENOME Human NPRC (NPR3) siRNA SMARTpool or siGENOME Non-Targeting siRNA pool (Horizon Discovery Biosciences; Cambridge, UK) using DharmaFECT2 transfection reagent (Horizon Discovery Biosciences) according to the manufactures instructions.

### 4.5. Cell Surface Biotinylation

Cultured cells were washed twice with 1XPBS before 0.5 mg/mL biotin (Thermo Fisher Scientific) was added to the surface of the cells. The biotin was quenched with quenching solution (18.3 mg/mL L-lysine) and the cells were scraped in mammalian protein extraction reagent (MPER) (ThermoFisher Scientific; Waltham, MA, USA). Next, the cell lysates were incubated with streptavidin beads (ThermoFisher Scientific) and incubated for 2 h with end-over-end rocking. The biotinylated fraction was eluted from the beads using Laemmli’s sample buffer (Bio-Rad; Hercules, CA, USA) and DTT (ThermoFisher Scientific).

### 4.6. SDS-PAGE and Western Blotting

The Bicinchoninic acid protein assay (Thermo Fisher Scientific) was performed to determine protein concentration of cellular lysates. Fifty micrograms of total protein from cellular lysates was prepared in Laemmli sample buffer and then loaded onto 4–20% Tris·HCl polyacrylamide gels. The proteins were resolved on the Criterion electrophoresis system (Bio-Rad) before being electrically transferred to nitrocellulose blotting membranes (Thermo Fisher Scientific) using the Criterion transfer system (Bio-Rad). A solution of 5% nonfat milk in 1× Tris-buffered saline (TBS; Bio-Rad) was used to block the membranes for 1 h at room temperature. The membranes were then washed with 1× TBS and then incubated with primary antibodies (ENaCα [43], ENaCβ [43], ENaCγ [44], NPRC antibody [5] at a dilution of 1:1000 in 5% BSA 1× TBS *(wt/vol)* while on a rocker at 4 °C overnight. Membranes were washed with 1× TBS before being incubated for 1 h in secondary antibody solution containing horseradish peroxidase-conjugated goat anti-rabbit secondary antibody at a dilution of 1:3000 prepared in blocking solution. Membranes were washed with 1× TBS, incubated with SuperSignal Dura Chemiluminescent Substrate, and then imaged on a Bio-Rad ChemiDoc MP Imaging System with Image Lab Software Version 6.1.0 Build 7 (Bio-Rad).

### 4.7. Lipid Extraction

Lipids were extracted according to the Bligh and Dyer method [45]. Briefly, 50 µL of sample was mixed with 950 µL water, placed in 10-mL glass screw-capped tube and kept on ice for 10 min. A mixture of 2 mL methanol and 0.9 mL methylene chloride was added followed by vortexing for 30 s. Samples were spiked with 50 µL of 5× diluted SPLASH internal standard (Avanti Polar Lipids, Inc., Alabaster, AL, USA) and incubated at room temperature for 30 min. One mL water and 0.9 mL methylene chloride were added and samples were gently inverted 10 times and then centrifuged at 200× *g* for 10 min. The organic lower phase (methylene chloride) was carefully collected using a glass Pasteur pipette. Samples were re-extracted in 2 mL methylene chloride and centrifuged. The bottom layer was collected, combined with the first extract, concentrated to dryness under an N2 stream and reconstituted into 50 µL of ethanol before analysis.

### 4.8. LC-MS/MS Conditions

Lipid samples were analyzed using an ultra-high-performance liquid chromatography system (UHPLC, Shimadzu Co., Kyoto, Japan) coupled to a QTrap 6500 mass spectrometer (AB Sciex, Redwood Shores, CA, USA). Chromatographic separation of lipids was performed using XBridge Amide 3.5 µm, 4.6 × 150 mm column (Waters, Dublin, Ireland). A binary gradient was applied using acetonitrile: water with the ratio of 95:5 (*v*/*v*) and 50:50 (*v*/*v*) for mobile phase A and B, respectively. Both solvents contained 1 mM ammonium acetate and the pH of freshly prepared mobile phases was adjusted to 8.2. The linear gradient of solvent B increased to 6% in 6 min, and then reached to 25% within 4 min, 98% within 1 min and finally 100% within 2 min with the flow rate of 0.7 mL·min^−1^. After each separation, the column was flushed using mobile phase B at a flow rate of 1.5 mL·min^−1^ for 3 min. Mass Spectrometry was operated in scheduled MRM.

The settings of the electrospray ionization source were as follows: Declustering potential for positive and negative modes was set to 60 and 80, respectively. Collision energy was varied from 25–60. Other fixed parameters were entrance potential and collision cell exit potential which were set at 15. Ion spray voltage was kept at 4.5 kV and temperature was 300 °C. Each sample was injected twice as technical replicates. Cross contamination was avoided using the following three techniques: flushing column and tubing using a high flow rate of mobile phase B, extra washing of needle using 500 µL isopropanol and running blanks as samples throughout the procedure at set intervals.

### 4.9. Protein Kinase C Activity Assay

A PKC activity assay (Abcam; 139437) was performed according to the manufactures instructions in order to calculate relative PKC activity in hAoEC lysates after treating the cells with vehicle or cANF for 8 h.

### 4.10. Statistical Analysis

All data are reported as mean values ± SEM and were compared using either the Student’s *t*-test or a one-way ANOVA in SigmaPlot software Version 14.0 (Systat Software, San Jose, CA, USA). Differences with a *p* value of <0.05 were considered statistically significant.

## Figures and Tables

**Figure 1 ijms-23-13959-f001:**
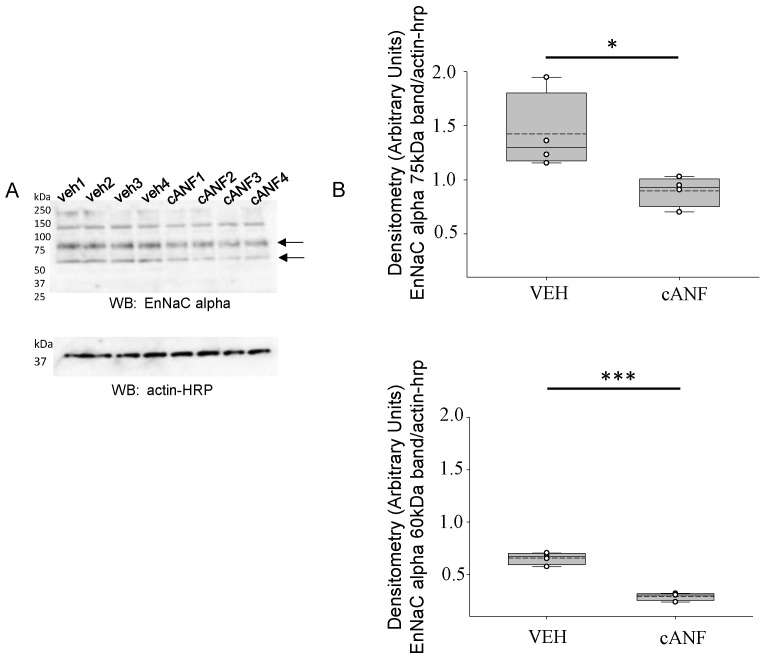
Western blot analysis of EnNaC alpha in human aortic endothelial cells. (**A**). Western blot of EnNaC alpha from cell lysates of 4 independents experiments in which hAoECs were treated with either vehicle (veh) or cANF for 8 h before harvesting the cells for protein. (**B**). Densitometric analysis of the immunoreactive bands indicated by arrows and normalized to the Actin band in the Western blot in (**A**). A Student’s *t* test was performed to compare the two groups. *n* = 4 samples in each of the two groups. * represents a *p*-value < 0.05. *** represents a *p*-value < 0.001.

**Figure 2 ijms-23-13959-f002:**
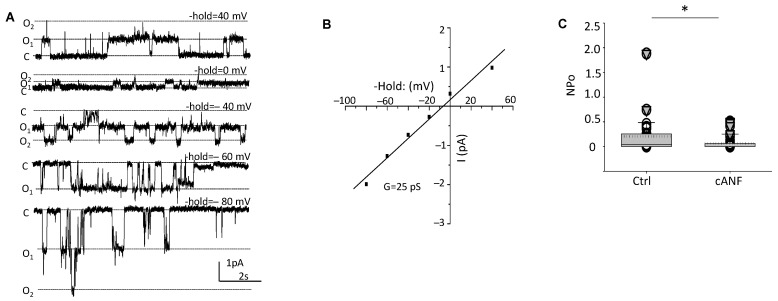
The NPRC agonist cANF inhibits EnNaC activity in hAoEC. Representative single-channel recordings (**A**), I-V curve of EnNaC (**B**), and summary plot of EnNaC activity (NPo) (**C**). A Student *t*-test was used to compare the control (Ctrl) and cANF experimental groups. *n* = 30 patches for the control group and *n* = 34 patches for the cANF group. * represents a *p*-value < 0.05.

**Figure 3 ijms-23-13959-f003:**
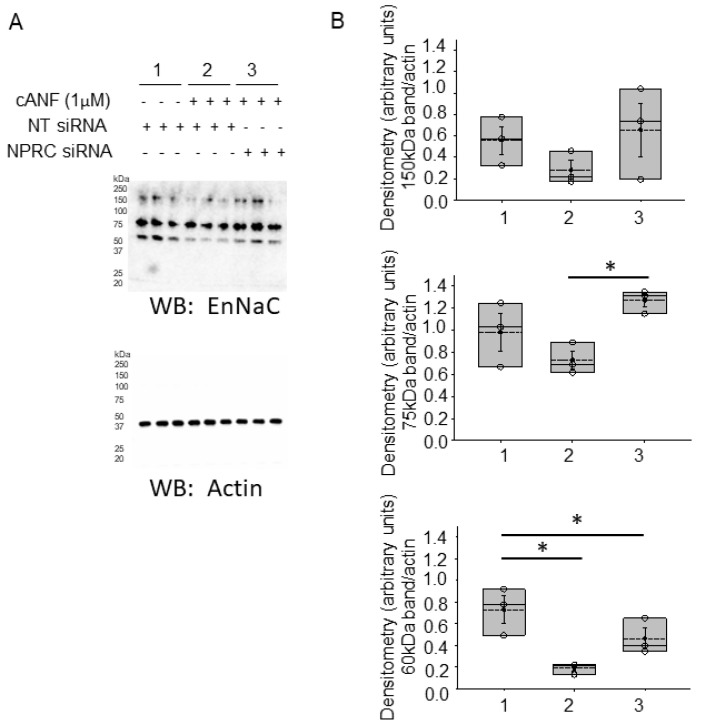
EnNaC alpha subunit protein expression in hAoEC transfected with NPRC specific siRNA compared to cells transfected with non-targeting siRNA. (**A**). Western blot for EnNaC alpha protein after hAoECs were transfected with NPRC specific siRNA or non-targeting (NT) siRNA and then treated with or without cANF before harvesting the cells for protein. (**B**). Densitometric analysis of the 150 kDa, 75 kDa, and 60 kDa immunoreactive bands in (**A**). Each group includes hAoEC cell lysates from three separate siRNA experiments that were loaded on the same gel and probed with EnNaC alpha antibody. A one-way ANOVA was used for statistical analysis of the experimental groups with a Holm–Sidak post hoc test. *n* = 3 samples for each of the three groups. * represents a *p*-value < 0.05.

**Figure 4 ijms-23-13959-f004:**
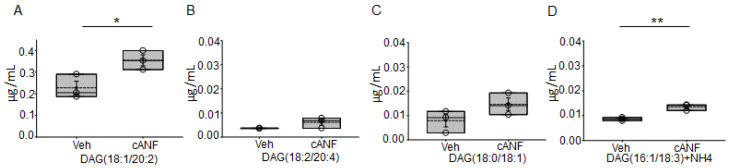
Quantification of diacylglycerols in hAoECs treated with cANF or vehicle. (**A**). DAG(18:1/20:2), (**B**). DAG(18:2/20:4), (**C**). DAG(18:0/18:1), (**D**). DAG(16:1/18:3) were among some diacylglycerols quantified in hAoECs treated with vehicle (Veh) or cANF for 8 h. A Student’s *t*-test was performed to make comparisons between the two groups. *n* = 3 samples in the vehicle and cANF groups. * represents a *p*-value < 0.05. ** represents a *p*-value < 0.01.

**Figure 5 ijms-23-13959-f005:**
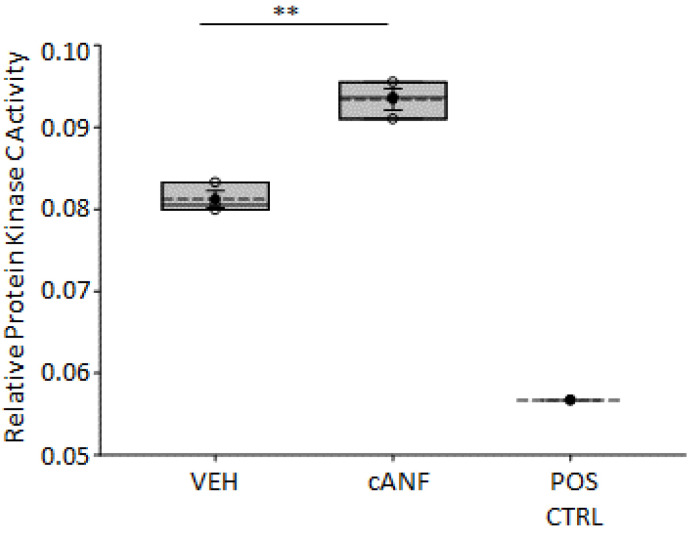
Protein Kinase C activity in hAoEC treated with cANF or vehicle. *n* = 3 independent cANF versus vehicle (VEH) control treatment experiments. POS CTRL represents Positive Control. A Student’s *t*-test was performed to make comparisons between the two groups. *n* = 3 samples for each of the two groups. ** Represents a *p*-value < 0.01.

**Figure 6 ijms-23-13959-f006:**
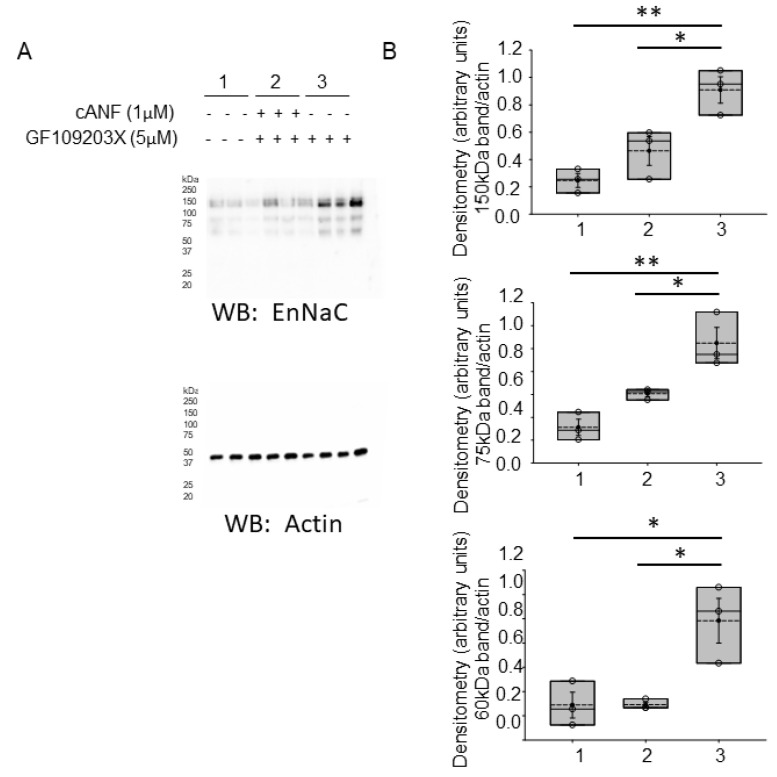
EnNaC alpha protein expression after inhibition of PKC.(**A**). Western blot for EnNaC alpha protein after hAoECs were treated with or without the PKC inhibitor GF109203X and with or without the NPRC agonist cANF for 3 h. (**B**). Densitometric analysis of the 150 kDa, 75 kDa, and 60 kDa immunoreactive bands in (**A**). A Student’s *t*-test was performed to make comparisons between the two groups. *n* = 3 samples for each of the three groups. * Represents a *p*-value < 0.05. ** Represents a *p*-value < 0.01.

**Figure 7 ijms-23-13959-f007:**
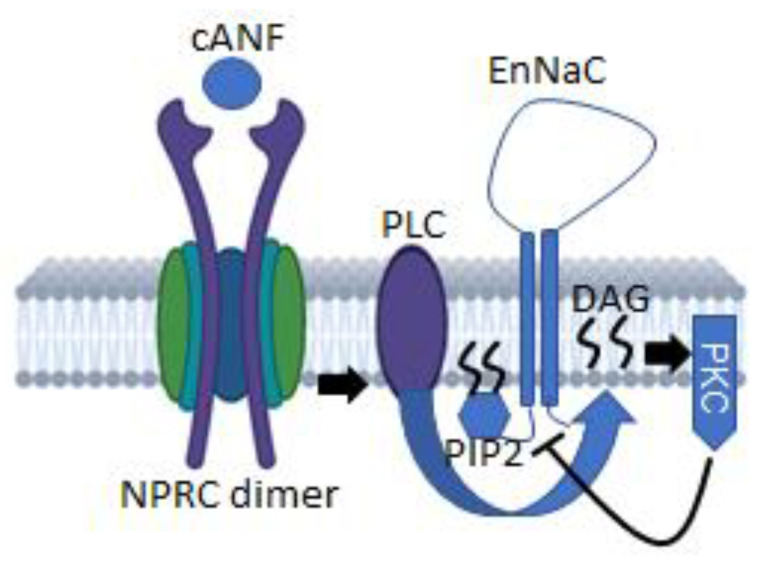
Schematic of the proposed model for the inhibition of EnNaC following NPRC activation. NPRC is activated by ligand binding. The NPRC agonist cANF is shown to selectively bind to NPRC. NPRC ligand biding results in the activation of phospholipase C (PLC) leading to the hydrolysis of PIP2. Hydrolysis of PIP2 produces the hydrophobic molecule diacylglycerol (DAG) which is retained in the plasma membrane and activates protein kinase C (PKC). PKC is known to inhibit the activity of ion channels including epithelial sodium channels (ENaC) and presumably endothelial sodium channels (EnNaC). Schematic was created in part using the BioRender software.

## Data Availability

The data from this study are presented within the figures. The lipidomic dataset is available at: https://doi.org/10.6084/m9.figshare.21586161.

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
