# Peer review of "C Type Natriuretic Peptide Receptor Activation Inhibits Sodium Channel Activity in Human Aortic Endothelial Cells by Activating the Diacylglycerol-Protein Kinase C Pathway"

_ijms, 2022, doi:10.3390/ijms232213959_

Round 1

Reviewer 1 Report

General comments

The authors examined whether the NPRC activation regulates EnNaC protein expression and its activity, and changes the DAG and PKC in hAoEC. The results obtained were consistent among experiments. This reviewer raises some issues in the manuscript writing and data reporting to improve the manuscript.

 Specific comments

Line 37 (Introduction): What is the potential relationship between the NPRC activation and endothelial cell stiffness? The authors explained a lot on the NPRC and EnNaC, however, there is no description on their hypothesis. Descriptions that are not related to the hypothesis generation are not necessary in the Introduction section.

 Line 63 (Results): In the last paragraph of the introduction section, the authors mentioned their first (i) and second (ii) goals. Why did the authors firstly show the results of their second goal? The results of the first goal should be shown first, then the results of second goal should be shown after that.

 Lines 65-67: In the Results section, the authors should not mention previous reports. They should mention previous reports in the Introduction section or Discussion section.

 Line 72: Readers can not recognize the number of plots in the figure. Please describe the number of samples (n) in the figure legend. There is the same issue in the other figures.

 Line 97: The word “increases” should be corrected to “decreases”.

 Lines 119-124: Font size is small.

 Line 156 (Discussion): This reviewer requests the authors to elaborate more on the potential association of the NPRC activation, the decreases in the EnNaC protein expression and its activation, and the increases in the DAG and PKC with the endothelial cell stiffness. Additionally, there is no conclusion statement at the end of the Discussion section. The conclusion must be succinct and definitely based on the present results.

Author Response

Reviewer 1

The authors examined whether the NPRC activation regulates EnNaC protein expression and its activity, and changes the DAG and PKC in hAoEC. The results obtained were consistent among experiments. This reviewer raises some issues in the manuscript writing and data reporting to improve the manuscript.

 Specific comments

Line 37 (Introduction): What is the potential relationship between the NPRC activation and endothelial cell stiffness? The authors explained a lot on the NPRC and EnNaC, however, there is no description on their hypothesis. Descriptions that are not related to the hypothesis generation are not necessary in the Introduction section. 

The potential relationship between NPRC activation and endothelial cell stiffness is now discussed in the fourth paragraph of the discussion section.  We added a description of our hypothesis in the first sentence of the last paragraph of the introduction section. 

 Line 63 (Results): In the last paragraph of the introduction section, the authors mentioned their first (i) and second (ii) goals. Why did the authors firstly show the results of their second goal? The results of the first goal should be shown first, then the results of second goal should be shown after that.

As suggested by the Reviewer, we have revised the order of the data to match the order of our goals stated in the introduction section of our manuscript.

 Lines 65-67: In the Results section, the authors should not mention previous reports. They should mention previous reports in the Introduction section or Discussion section. 

Previous reports have been removed from the results section as suggested by the Reviewer. 

 Line 72: Readers can not recognize the number of plots in the figure. Please describe the number of samples (n) in the figure legend. There is the same issue in the other figures. 

The number of samples in each group is now stated in the figure legends for all figures in the manuscript.

 Line 97: The word “increases” should be corrected to “decreases”. 

Increases was corrected to decreases

 Lines 119-124: Font size is small. 

The extra figure legend was removed in lines 119-124.

Line 156 (Discussion): This reviewer requests the authors to elaborate more on the potential association of the NPRC activation, the decreases in the EnNaC protein expression and its activation, and the increases in the DAG and PKC with the endothelial cell stiffness. Additionally, there is no conclusion statement at the end of the Discussion section. The conclusion must be succinct and definitely based on the present results. 

As suggested by the Reviewer, we have expanded our discussion on the potential association between NPRC activation and the resulting decrease in EnNaC protein expression and activation, in a mechanism involving DAG and PKC.  Please see the fourth paragraph of the discussion section, which we added to our revised manuscript.  Also, we added the fifth sentence of the fifth paragraph of the discussion section to reflect the significance of our newly added data on the pharmacological inhibition of PKC.   Finally, a conclusion paragraph was added the end of the discussion section.

Reviewer 2 Report

The manuscript by Yu et al. showed that the activation of the natriuretic receptor (NPRC) results in decreased EnNaC activity.

8 hour stimulation of NPRC resulted in lower amount of alpha-EnNaC protein (whole cell lysates and biotinylated fraction) and higher levels of certain diacylglycerols. Interestingly, alpha-EnNaC protein levels were not decreased in endothelial cells lacking NPRCs.

NPRC stimulation resulted in decreased EnNaC activity. However, functional experiments were performed on endothelial cells stimulated for 15 minutes, which strongly differs from the biochemical experiments where NPRC was stimulated for 8 hrs.

NPRC stimulation activated PKC, which was proposed to act as a mediator in the molecular mechanism involved in EnNaC inhibition by NPRC stimulation. Unfortunately, no experiments linking PKC and EnNaC inhibition upon NPRC stimulation was presented in the current study.

In summary, the manuscript is well written but there are major issues that need to be addressed.

I recommend its acceptance if the following issues are addressed.

Major issues:

1) Perform biochemical and functional experiments with the same NPRC stimulation time.

2) Evaluate the involvement of PKC in the NPRC-mediated EnNaC inhibition. An easier way to evaluate this is to evaluate at the protein level (or channel activity), the NPRC-mediated EnNaC inhibition in the presence of PKC inhibitors.

Minor issues.

1) Remove the extra figure legend in lines 119-124

2) Remove immunocytochemistry protocol in the methods section as no localization study is included in the current version of the manuscript

Author Response

The manuscript by Yu et al. showed that the activation of the natriuretic receptor (NPRC) results in decreased EnNaC activity.

8 hour stimulation of NPRC resulted in lower amount of alpha-EnNaC protein (whole cell lysates and biotinylated fraction) and higher levels of certain diacylglycerols. Interestingly, alpha-EnNaC protein levels were not decreased in endothelial cells lacking NPRCs.

NPRC stimulation resulted in decreased EnNaC activity. However, functional experiments were performed on endothelial cells stimulated for 15 minutes, which strongly differs from the biochemical experiments where NPRC was stimulated for 8 hrs.

NPRC stimulation activated PKC, which was proposed to act as a mediator in the molecular mechanism involved in EnNaC inhibition by NPRC stimulation. Unfortunately, no experiments linking PKC and EnNaC inhibition upon NPRC stimulation was presented in the current study.

In summary, the manuscript is well written but there are major issues that need to be addressed.

I recommend its acceptance if the following issues are addressed.

Major issues:

1) Perform biochemical and functional experiments with the same NPRC stimulation time. 

We acknowledge this important point by the Reviewer.  Typically, we are able to see changes in EnNaC activity after treating cells with an agonist or drug for a relatively short period of time.  Changes in EnNaC protein expression measured by Western blotting is more difficult to observe since EnNaC is a rare protein and that is the reason we initially chose an 8 hour time point for our protein biochemistry experiments.  We corrected our methods section to reflect that we measured EnNaC activity in a window from 30 minutes to 3 hours after cANF treatment (instead of 15 minutes as we incorrectly stated in the methods section).  The time it took to acquire a patch varied from cell to cell so we now provide a time range for the electrophysiology experiments. Nevertheless, as suggested by the Reviewer, we performed additional Western blot experiments with a similar NPRC stimulation time (3 hours) used in our patch clamp studies. This new data is shown in the new Figure 3.  

2) Evaluate the involvement of PKC in the NPRC-mediated EnNaC inhibition. An easier way to evaluate this is to evaluate at the protein level (or channel activity), the NPRC-mediated EnNaC inhibition in the presence of PKC inhibitors. 

As suggested by the Reviewer, we performed additional experiments using the PKC inhibitor GF109203X to further show the involvement of PKC in the NPRC-mediated inhibition of EnNaC. Please see this additional data in the new figure 6. 

Minor issues.

1) Remove the extra figure legend in lines 119-124  

The extra figure legend has now been removed.

2) Remove immunocytochemistry protocol in the methods section as no localization study is included in the current version of the manuscript. 

The immunocytochemistry protocol has been removed from the methods section. 

Round 2

Reviewer 2 Report

The revised version of the manuscript includes new experiments that address the concerns raised before.

I recommend the acceptance of this paper after addressing the following minor issues:

Figure 2:

The Po values are quite low and the estimation of channel number in each patch might be underestimated. Therefore, I recommend to remove figs 2d and e.

Please correct the legend that contains mistakes.

Typos:

Line 200

Line 203, correct figure number (should be 3)

Line 212, replace activity by protein levels as western blots (not single channel recordings) were performed.

Author Response

Figure 2:

The Po values are quite low and the estimation of channel number in each patch might be underestimated. Therefore, I recommend to remove figs 2d and e.

As recommended, we have removed Figs 2d and e

Please correct the legend that contains mistakes.

The legend has now been corrected

Typos:

Line 200

The typo on Line 200 has been corrected

Line 203, correct figure number (should be 3)

The typo on line 203 has been corrected

Line 212, replace activity by protein levels as western blots (not single channel recordings) were performed.

Line 212 has been corrected as suggested